# ACE-031, a soluble activin type IIB receptor, increases muscle mass and strength in the common marmoset (Callithrix jacchus)

**Samuel M. Cadena**[1☉], **Sasha Bogdanovich**[2☉], **Tejvir S. Khurana**[2], **Abigail Pullen**[1],
**R. Scott Pearsall**[1], **Elizabeth Curran**[3], **Ryan Faucette**[1], **Joan Lane**[1], **Jasbir Seehra**[1],
**Jennifer L. Lachey**[1], **Alan D. Mizener**[4], **Emidio E. Pistilli**[2¤] *

**1** Acceleron Pharma Inc., Cambridge, Massachusetts, United States of America, **2** Department of Physiology and Pennsylvania Muscle Institute, University of Pennsylvania School of Medicine, Philadelphia, Pennsylvania, United States of America, **3** Harvard Medical School/New England Regional Primate Research Center, Southborough, Massachusetts, United States of America, **4** West Virginia University Cancer Institute, West Virginia University School of Medicine, Morgantown, West Virginia, United States of America

☉ These authors contributed equally to this work.
¤ Current address: Division of Exercise Physiology and West Virginia University Cancer Institute, West Virginia University School of Medicine, Morgantown, West Virginia, United States of America
* epistilli2@hsc.wvu.edu

## Abstract

Pharmacological blockade of ligands for the activin receptor type IIB (ActRIIB) e.g., myostatin and activin A is associated with improvements in murine skeletal muscle mass and function. The efficacy of a similar treatment approach in a non-human primate (NHP) model would suggest a greater likelihood of success in the treatment of humans suffering from chronic myopathies. In the present study, we elucidate the potential therapeutic benefit of ACE-031, a therapeutic protein consisting of the ActRIIB extracellular region fused to human IgG1, in the common marmoset (Callithrix jacchus). Marmosets were randomized to receive ACE-031 or vehicle control (10 mM Tris buffered saline; TBS) for 14 weeks. Body composition was measured weekly throughout the experimental period and morphometric analysis and contractile properties of skeletal muscle were assessed terminally. There was a significant main effect of time and time x treatment interaction for lean body mass, such that marmosets administered ACE-031 were greater at euthanasia compared to baseline; this was not observed in the vehicle-treated controls. Biceps brachii exhibited a significant increase in the cross-sectional area of both type I and type II fibers and ex vivo contractile properties of the EDL showed an increase in absolute and specific force production. The efficacy of ACE-031 in non-human primates provides optimism that a therapeutic strategy that targets multiple negative regulators of skeletal muscle may be beneficial in treating myopathies in humans.

**Data availability statement:** All relevant data are within the manuscript and its Supporting information files.

**Funding:** The author(s) received no specific funding for this work.

**Competing interests:** I have read the journal's policy and the authors have declared no competing interests. Co-authors were former employees of a company Acceleron that holds the patent to the drug. However, the company is no longer active. This does not alter our adherence to PLOS ONE policies on sharing data and materials.

## Introduction

Members of the transforming growth factor beta (TGF-β) superfamily, including myostatin (GDF-8) and activin A, are important regulators of skeletal muscle growth [1,2]. Overexpression of myostatin or activin A is associated with muscle wasting in both human chronic disease [3–7] and animal models [8–11]. In contrast, deficiency of either or both ligands in the form of genetic deletion [12–17] or pharmacological inhibition [17–22] results in increased muscle mass and rescue of muscle loss in models of muscle wasting. These findings suggest that therapeutics specifically targeting myostatin or activin A signaling to stimulate skeletal muscle growth may provide therapeutic benefit in human myopathies and conditions associated with muscle wasting.

In several mouse models of muscular dystrophy, myostatin inhibition is associated with modest improvements in dystrophic muscle, including increased muscle mass and fiber cross-sectional area, improved functional capacity and reduced degeneration and fibrosis [18,19,23–27]. However, when this same therapeutic strategy was applied clinically to patients afflicted with muscular dystrophies, the outcomes were not as promising [28–31]. While some improvements in force production were observed in a single muscle fiber preparation [28], investigators failed to observe any meaningful clinical benefit in patients [28–31]. It has been reported that circulating activin A levels in monkeys and humans are 3–4-fold higher than mice and rats [21], suggesting that myostatin specific treatments that appear promising in mice are not likely to be as impactful in humans. Thus, a more comprehensive treatment modality that targets both activin A and myostatin may be necessary to effectively counter the degeneration, weakness and functional decline of muscle associated with these myopathies in humans.

Myostatin and activin A signal through a heteromeric complex that consists of type I and II receptors of the TGF-β superfamily. Deletion or inhibition of the activin type IIB (ActRIIB) receptor in adult mice is associated with significant increases in skeletal muscle mass [32–36] that are greater than inhibition of myostatin alone [21,37], suggesting that the simultaneous inhibition of both myostatin and activin A could potentially present a more effective therapeutic strategy in the treatment of muscle disorders. Indeed, in a phase I trial evaluating anti-myostatin and anti-activin A antibodies in post-menopausal women, the dual antibody treatment was superior to either antibody alone in their ability to increase thigh muscle volume and total lean body mass [38]. Similarly, treatment with both antibodies significantly increased muscle mass in mice and non-human primates subjected to combined diet induced obesity and GLP-1-induced muscle loss [39]. Utilizing a ligand trap based treatment modality to inhibit myostatin and activin A with a single molecule, post-menopausal women treated with one dose of ACE-031 (a soluble form of the ActRIIB extracellular domain fused to the Fc portion of human IgG) exhibited a 3.3% increase in lean body mass and a 5.1% increase in thigh muscle volume, as assessed by DEXA and MRI, respectively [40]. In a follow-up study in boys with Duchenne muscular dystrophy, ACE-031 showed trends for increased lean body mass [41]. However, the study was terminated prematurely after the second dosing regimen due to safety concerns

around epistaxis and telangiectasias. Later reports indicate that bleeding events were likely due to inhibition of BMP9, a ligand critical for vascular remodeling [42]. Thus, despite showing a tendency to increase muscle mass and prevent decline in the 6-minute walk time [41] the full clinical benefit of ACE-031 was never fully realized. Therefore, to better understand the full therapeutic potential of ACE-031 over a longer treatment duration and to bolster confidence that a similar treatment modality (i.e., a ligand trap inhibiting both myostatin and activin A) is likely to confer significant clinical benefit in muscular disorders, we herein report results from a 14-week dosing study in the common marmoset, characterizing the effects of ACE-031 on body weight, muscle mass, muscle fiber morphology and isometric strength.

## Materials and methods

### Expression and purification of ACE-031

The ActRIIB extracellular domain was produced by PCR amplification of the human *ACVR2B* gene. The primers used for the 5′ and 3′ ends included a SfoI and AgeI restriction site, respectively. PCR product was purified, digested with SfoI and AgeI and ligated into the pAID4 hFc vector (containing the tissue plasminogen activator signal sequence) to create the pAID4 ActRIIB.hFc expression construct. The sequences of ActRIIB extracellular domain and human immunoglobulin Fc region were confirmed by double strand dideoxy sequencing. ActRIIB.hFc plasmid was transfected into Chinese Hamster ovary (CHO) cells and a stable clone expressing ActRIIB.hFc was isolated. ActRIIB.hFc was purified from serum free media using affinity chromatography with Mab Select Sure Protein A (GE Healthcare, Piscataway NJ) and additional purification steps as needed. Purified ActRIIB.hFc was dialyzed into tris buffered saline (TBS; 10 mM tris, 137 mM NaCl and 2.7 mM KCl, pH 7.2).

### Animals

Six male and six female healthy adult common marmosets (Callithrix jacchus) were included in this study. Animals were part of The New England Primate Research Center breeding colony and were maintained in accordance with the "Guide for the Care and Use of Laboratory Animals" of the Institute of Laboratory Resources, National Research Council. The facility is accredited by the Association for the Assessment and Accreditation of Laboratory Animal Care International and all work was approved by Harvard Medical School's Standing Committee on Animals. Animals were randomized to control (n = 4) or treatment groups (n = 8), with an equal number of male and female animals in each group. Animals were dosed once weekly for 14 weeks by interscapular subcutaneous injection at a dose of 3.0 mg·kg$^{-1}$ of ACE-031 or an equal volume of vehicle as a control (10 mM Tris buffered saline; TBS). Animals were euthanized at week 14 for tissue collection. Euthanasia was by pentobarbital overdose delivered intravenously under ketamine sedation.

### Body composition

Measurements were obtained in conscious animals at baseline and every two weeks throughout dosing. Animals were habituated to entering an elongated holder for a food reward. Total body weight was obtained by weighing the holder and animal on an electronic scale. Fat mass and lean body mass were then quantified using an EchoMRI quantitative magnetic resonance analyzer (QMR; Echo Medical Systems, Houston, TX). The QMR analyzer was calibrated daily prior to analysis with a standard consisting of a measured amount of canola oil. Animals were measured three times during each session with the means used for further analyses.

### Blood sampling

Blood collection for complete blood cell and serum chemistry analysis was performed under ketamine sedation (15 mg·kg$^{-1}$ Ketaset IM; Fort Dodge, Overland Park, KS) at baseline and weeks 1, 4 and 8 following dosing. Terminal blood samples were taken at the time of necropsy.

### Muscle fiber cross-sectional area (CSA) and fiber type distribution

Immediately following euthanasia, biceps brachii muscles were harvested, the long head was bisected near the mid-belly region, snap frozen in liquid-nitrogen-cooled isopentane and stored at −80 °C for subsequent analyses. Ten μm sections were fixed on glass slides with 10% formalin for 20 min. For CSA analyses, sections were incubated in wheat germ agglutinin (WGA) conjugated to Alexa Fluor 488 diluted in PBS (Invitrogen, Carlsbad, CA) for 60 min at room temperature. To distinguish between fiber types, sections were incubated in anti-fast (type II) MHC conjugated to alkaline phosphatase (Sigma, Saint Louis, MO) diluted in PBS for 60 min at room temperature then incubated for 10 minutes in Vector® Red Alkaline Phosphatase Substrate Kit (Vector Laboratories, Burlingame, CA). Stained sections were visualized with an Eclipse 80i fluorescent microscope and images captured with a Digital Sight DS-5Mc digital camera (Nikon Corporation, Melville, NY). Images were analyzed with NIS Elements imaging software (Nikon Corporation). For each sample an average of 942 fibers per section were measured to determine cross-sectional area and all fibers per section were counted to calculate fiber type distribution.

### Ex vivo contractile properties

EDL muscles were dissected and tied around the proximal and distal tendons of the muscle with 6−0 sutures. The muscles were transferred to a stimulation chamber and muscle length was adjusted to achieve the maximal twitch response ($L_o$). Three twitch contractions were performed with muscle at $L_o$. Maintaining the muscle at $L_o$, the force-frequency relationship was determined. Tetanic contractions were performed at 30 Hz, 50 Hz, 80 Hz, 100 Hz, and 120 Hz and the maximal force plotted for each frequency. The maximal forces occurred at a stimulation frequency of 50 Hz, therefore, further analyses were performed at 50 Hz. Three tetanic contractions (50 Hz, 500ms) were performed with the muscle at $L_o$. All three twitches and tetani were analyzed and the maximal responses were averaged for each group.

### Statistical analysis

Data are reported as means ± standard error of the mean unless stated otherwise. Non-normally distributed variables were log-transformed so that data were normally distributed prior to statistical analysis. Muscle weights, contractile properties and anthropometric measurements at euthanasia were analyzed using two-tailed unpaired Student's t-test for between-group comparisons. Bi-weekly measures of body weight and lean body mass were analyzed using a 2-way repeated measures ANOVA and Bonferroni's multiple comparisons post-hoc test. Absolute forces obtained in the force-frequency relationship were analyzed using a 2-way ANOVA and Bonferroni's multiple comparisons post-hoc test. Significance was set at $p < 0.05$.

## Results

### Safety and tolerability, body weight and lean body mass

Marmosets received weekly interscapular subcutaneous injections of ACE-031 at a dose of 3.0 mg· kg$^{-1}$ for 14 weeks. The drug was generally well tolerated, and marmosets showed no adverse effects associated with treatment. Blood samples were collected at several intervals throughout the course of the study to evaluate serum chemistries and all values remained within normal ranges (S1 Table). Upon necropsy, no gross safety concerns or morphological abnormalities in tissues or organs were observed. The 2-way ANOVA identified a main effect of time ($p = 0.0092$) for absolute body weight changes during the 14 weeks. However, the main effect of treatment ($p = 0.6762$) and the time x treatment interaction ($p = 0.4406$) were not significant. Compared to baseline, the body weight in ACE-031 treated marmosets were significantly greater at weeks 4–12. In contrast, compared to baseline the vehicle control group, treated with an equal volume of TBS, showed no significant differences in body weight (Fig 1A). The 2-way ANOVA identified a significant main effect of time ($p < 0.0001$), and a time x treatment interaction ($p = 0.0012$) for absolute changes in lean body mass during the 14 weeks.

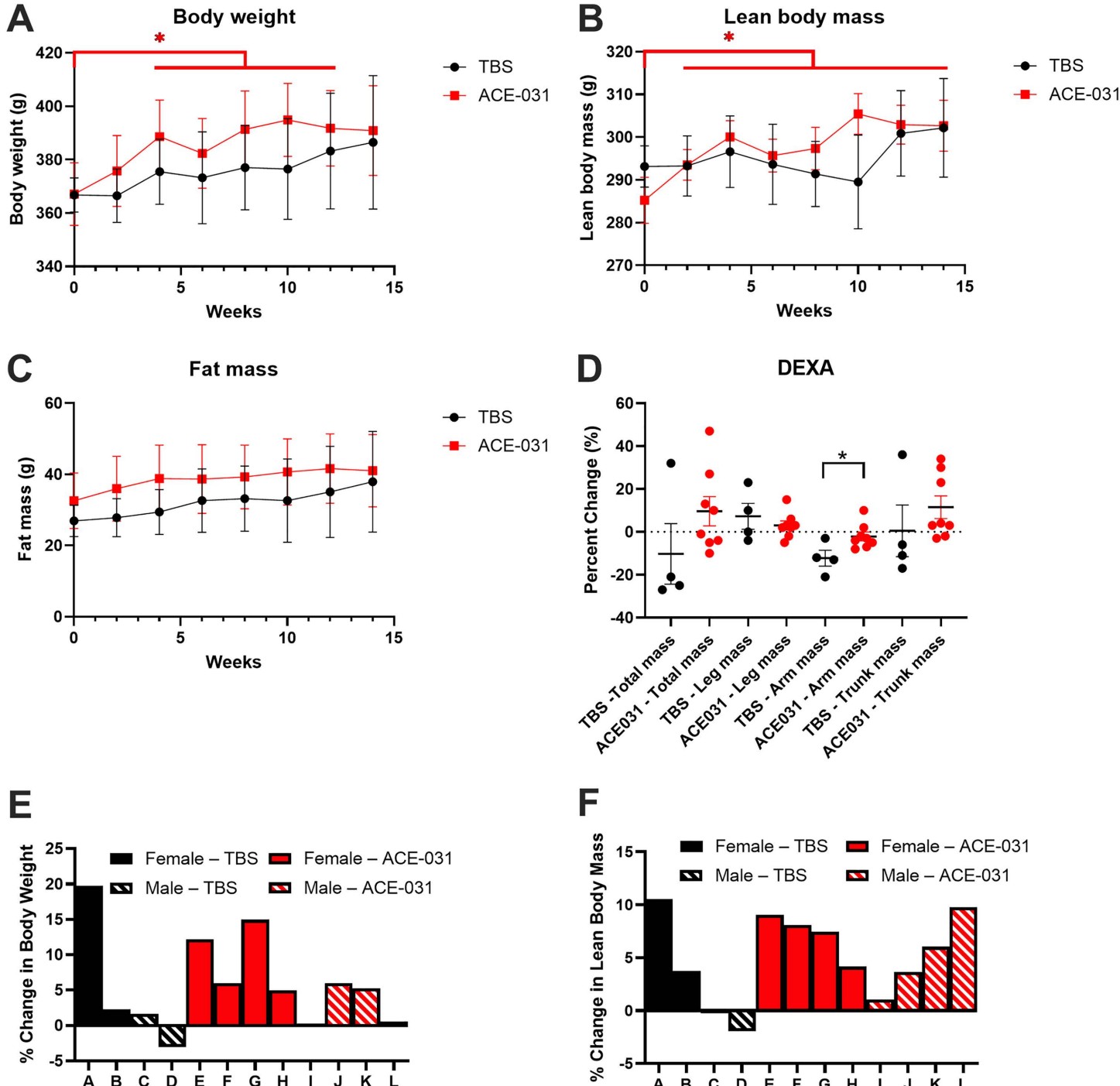

**Fig 1. Body weight and body composition analysis. (A)** Body weights of ACE-031 treated marmosets and control at the end of the treatment compared to baseline during the treatment period (week 0 versus week 14). **(B)** Lean body mass of ACE-031 treated marmosets was significantly greater at the end of the treatment (week 14) when compared to its baseline (week 0) while no differences were observed in control during the treatment period (week 0 versus week 14). **(C)** No differences in fat mass were observed in either group during the treatment period. **(D)** There was a significant difference in the muscle mass of the arms when comparing ACE-031 treated marmosets to control marmosets at the end of the treatment period. **(E)** The percent change in body weight for individual marmosets in the over the duration of the treatment period. **(F)** The percent change in body weight for individual marmosets in the over the duration of the treatment period.

The time x treatment interaction suggests that lean body mass in ACE-031 treated marmosets was greater over time while lean body mass remained stable in control marmosets. However, the main effect of treatment (p = 0.7506) was not significant. Compared to baseline, lean body mass, as assessed by EchoMRI, in ACE-031 treated marmosets was significantly greater at weeks 2–14 (Fig 1B). There were no differences in fat mass over the 14 weeks of treatment (Fig 1C). Muscle mass analyzed using DEXA showed a significant difference in lean body mass of the arms of ACE-031 treated animals compared to controls (p = 0.0225) (Fig 1D). The same observation was not confirmed for total body mass or in other muscle compartments such as the trunk and legs.

The percent change in body weight and lean body mass of individual marmosets are depicted in Fig 1E and 1F. The TBS-treated marmoset labeled as "A" presented with a relatively large increase in both body weight and lean body mass by the end of the experiment. Blood chemistries for this marmoset were all within normal ranges (S1 Table) and gross analysis upon necropsy revealed no apparent abnormalities that would explain a large degree of weight gain. In addition, pharmacokinetic assays confirmed that ACE-031 was not present in the serum of this marmoset (S2 Table), indicating the accumulation of body weight and lean body mass was not due to inadvertent ACE-031 exposure. We did not exclude this marmoset from statistical analyses and suggest that the overall change in body weight of this specific marmoset may have affected the group means and therefore the statistical comparisons between groups. Blood chemistries for all marmosets were within normal ranges (S1 and S2 Tables).

## Muscle fiber CSA and fiber-type distribution

To further explore the anabolic propensity of ACE-031 in skeletal muscle and to corroborate changes in arm muscle mass observed by DEXA, a morphometric analysis of the biceps brachii was performed and muscle fibers were examined for differences in CSA and myosin heavy chain (MHC) isoform expression. Representative muscle images stained for MHC type II are shown in Fig 2A and 2B. Type II muscle fiber CSA was 20% greater and type I muscle fiber CSA was 34% greater in ACE-031 treated marmosets compared to TBS treated marmosets (Fig 2C and 2D). The type II and type I fiber CSA distributions in ACE-031 treated marmosets were shifted to the right toward larger muscle fiber CSA (Fig 2E and 2F). It is not known whether inhibition of ActRIIB signaling in skeletal muscle induces shifts in muscle fiber type. Therefore, the proportion of type I fibers per muscle section of the biceps brachii was determined. Biceps brachii muscles of both TBS- and ACE-031-treated marmosets consisted of 16% type I fibers (S1 File). Thus, up to a 14-week blockade of ActRIIB ligands does not alter the MHC fiber type profile of skeletal muscle in the common marmoset.

## Ex vivo contractile properties

To determine if ACE-031 treatment is associated with improvements in muscle function, we measured *ex vivo* contractile properties of the EDL. While absolute twitch force was not significantly different, specific twitch forces were significantly greater in EDL muscles from ACE-031 treated animals (Fig 3A and 3B). As the contractile properties of marmoset skeletal muscle have not been previously characterized, we established a force-frequency relationship for the EDL muscles to determine the stimulation frequency for optimal force production (Fig 3C). In both treatment groups, the greatest force was generated in response to a 50 Hz stimulus; therefore, the 50 Hz stimulation frequency was used to obtain maximal isometric tetanic force. While absolute tetanic isometric force was not significantly greater, specific tetanic force was greater in EDL muscles from ACE-031 treated animals (Fig 3D and 3E). Collectively, these data indicate an improvement in isometric muscle strength that is suggestive of enhanced muscle function.

## Discussion

Myostatin has been shown to induce atrophy by inhibiting signaling pathways associated with protein synthesis and activating those associated with degradation [43–46]. Whereas, inhibition of myostatin and activin A has been shown to preserve muscle mass by preventing perturbations in these pathways [46]. Activin A levels in monkeys and humans

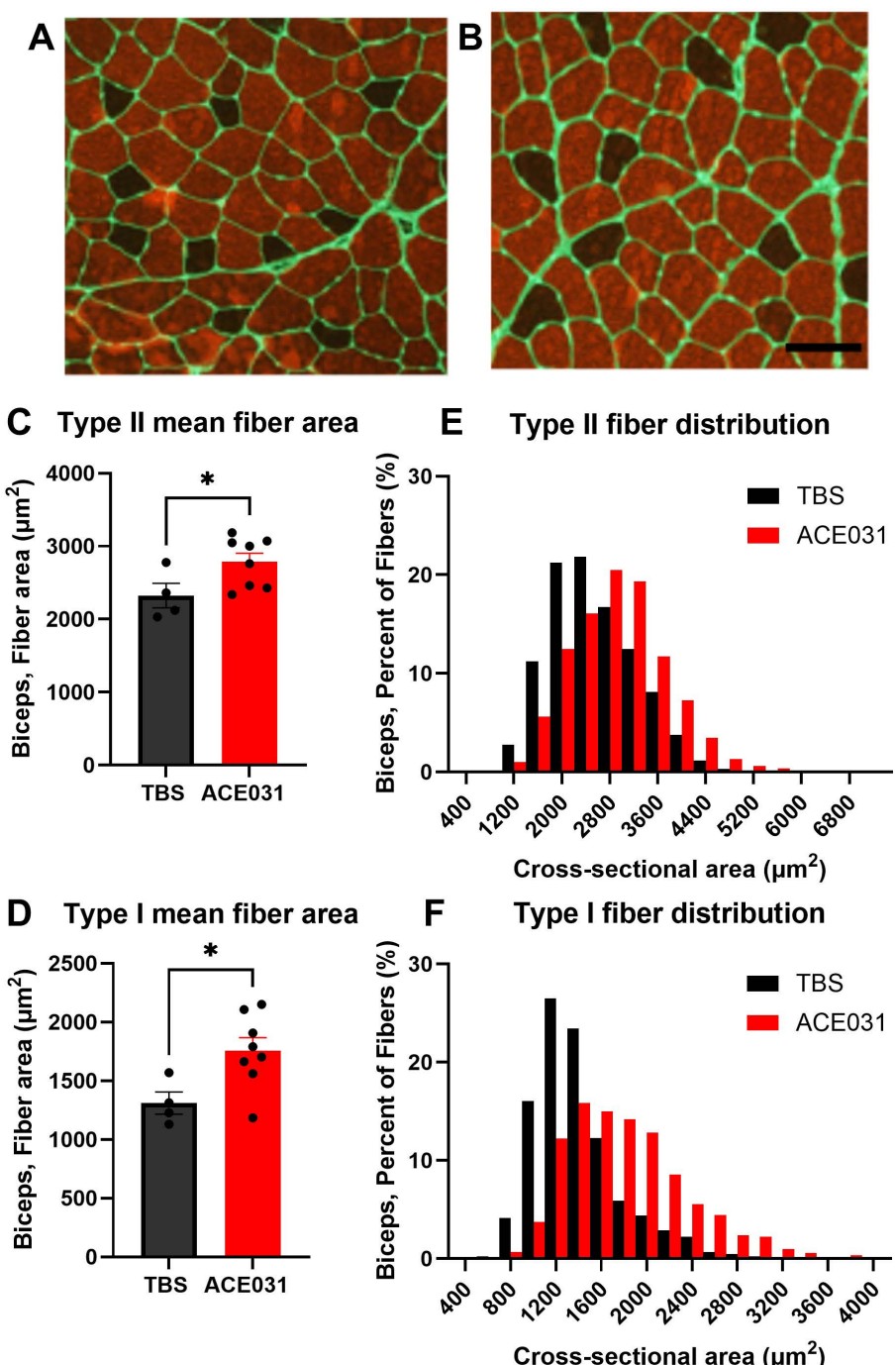

**Fig 2. Biceps brachii fiber type analysis. (A, B)** Representative biceps brachii muscle sections from control and ACE-031 treated marmosets. **(C, D)** Type II fibers (fast) and type I fibers (slow) had significantly increased cross-sectional area. **(E, F)** Type II fibers and type I fibers exhibited a rightward shift of the fiber CSA distribution with ACE-031 treatment.

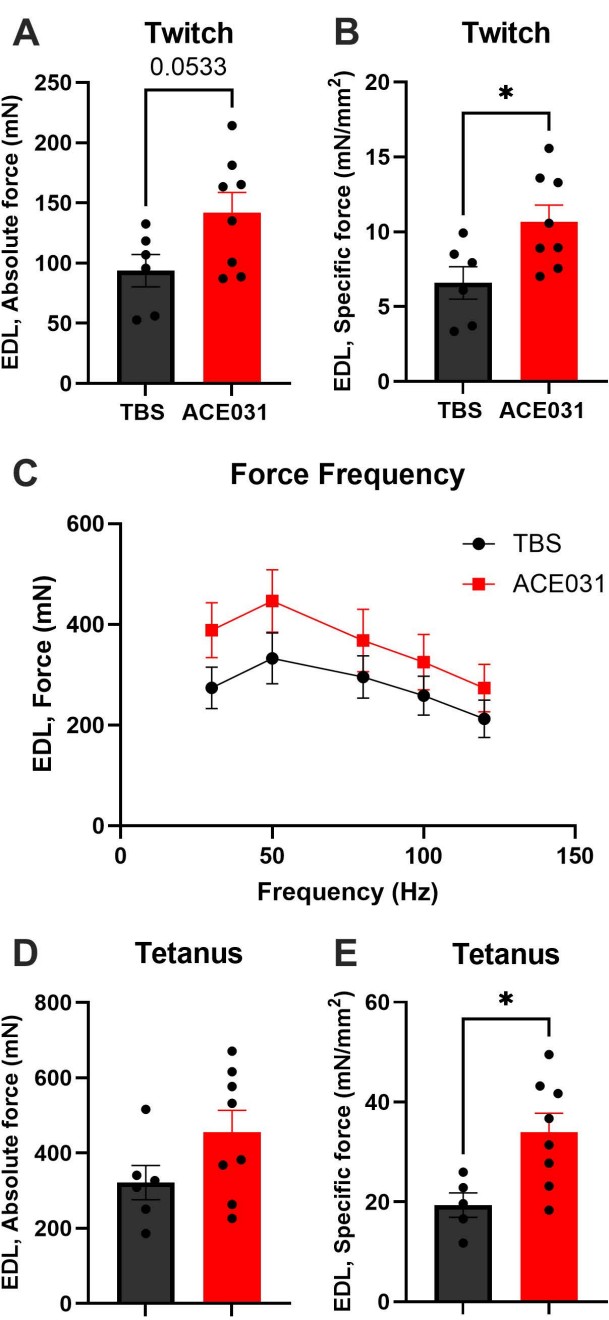

**Fig 3. Ex vivo muscle functional assessment. (A)** Absolute isometric twitch force of the EDL in ACE-031 treated marmosets compared to control marmosets. **(B)** Specific isometric twitch force of the EDL was significantly greater in ACE-031 treated marmosets compared to control marmosets. **(C)** The optimal stimulation frequency of the EDL was determined from the Force-Frequency relationship curve. **(D)** Absolute isometric tetanic force of the EDL was not significantly different when comparing ACE-031 treated to control marmosets. **(E)** Specific isometric tetanic force of the EDL was significantly greater in ACE-031 treated marmosets compared to control marmosets.

are 3–4-fold higher than mice and rats [21] and myostatin levels in mice are more than 10-fold higher than humans [47] suggesting that myostatin specific treatments that appear promising in mice are not as likely to produce clinically meaningful outcomes in humans. Indeed, although myostatin specific treatments have been shown to be efficacious in a mouse model of muscular dystrophy [18,24,27], selective antagonism of myostatin has failed to show meaningful clinical benefit in patients with muscular dystrophies [29–31]. Further, a systematic analysis of the pharmacokinetic and pharmacodynamic profile of a myostatin neutralizing antibody in mice, rats, monkeys and humans suggest that the amount of drug needed to induce comparable increases in muscle mass in monkeys was 20-fold higher than what is needed in mice; indicating substantial species differences in sensitivity to myostatin specific treatments [48]. Thus, the use of NHPs is likely to be a superior predictor of efficacy for treatments targeting myostatin and activin A in humans.

Current standard drug therapies for muscular dystrophies predominantly rely on the use of corticosteroids [49]. While these treatments have been shown to slow disease progression by reducing inflammation and cell death, chronic corticosteroid use is associated with several side effects. These include behavioral abnormalities [50], increases in fat mass [51] and insulin resistance [52] and decreases in bone mineral density [53]. Recently, several novel treatments based on exon skipping and gene therapy have been approved. However, they are predicted to benefit only a small segment of the muscular dystrophy patient population and, in the case of the first approved gene therapy, have the possibility of serious adverse effects such as liver and heart injury [54–56]. Thus muscular dystrophy patients are in need of a new therapeutic approach that has the potential to benefit all patients without the risk of serious side effects.

Here we have described the effects of the pharmacological inhibition of ActRIIB ligands in the common marmoset by administration of ACE-031, a soluble form of ActRIIB fused to the Fc portion of human IgG. Administration of ACE-031 for 14 weeks produced a significant increase in marmoset body weight from baseline that was paralleled by an increase in lean body mass; suggesting that greater body mass associated with loss of ActRIIB-signaling is likely the result of increased skeletal muscle growth. This is consistent with several studies investigating myostatin inhibition alone [18,20,57] or in combination with inhibition of activin A [39,58–60], including the administration of a murine analog of ACE-031 (RAP-031, the extracellular region of ActRIIB fused to the Fc region of a murine IgG2) [32,33,59,61,62] and a monoclonal antibody against ActRIIA and ActRIIB [63,64] in models of muscle wasting and degeneration. In the current study, total fat mass did not differ between ACE-031 and TBS treated marmosets. This is consistent with previous reports showing that as opposed to obese mice, lean mice have a reduced tendency to shed fat mass when treated with ActRIIB-Fc [65]. The absence of changes in fat mass in this study further corroborates that the increase in body weight was the result of an increase in lean body mass.

Genetic deletion of myostatin is associated with a greater proportion of fast glycolytic fibers at the expense of slow oxidative fibers [66–68]; an effect that also persists in heterozygous myostatin KO animals [69]. This appears to be purely a developmental effect as mice administered an AAV expressing a dominant negative form of myostatin present with an increase in IIB fibers when treated as neonates but not when treated as adults [70]. Girgenrath et al., also showed that treatment of mature mice with a myostatin specific antibody for 12 weeks did not impact fiber-type distribution [68]. Similarly, we have shown here in the common marmoset and previously in mice that fiber type distribution is not altered by ACE-031 induced hypertrophy [71]. We have also shown here and previously [71] that both type I and II fibers respond to ACE-031 induced hypertrophy, exhibiting comparable increase in fiber cross-sectional area. Thus, it is expected that the inherent metabolic profiles of the muscle (i.e., fast/glycolytic vs slow/oxidative) are likely to remain intact and unaltered by ACE-031 treatment. This contrasts with other groups reporting that myostatin inhibition induced hypertrophy shows a clear selectivity for type IIB fibers [57,72,73]. Slow-twitch fibers maintain oxidative capacity and fatigue resistance, supporting posture and endurance-based tasks, whereas fast-twitch fibers provide high power output and rapid contractile kinetics needed for short bursts of energy. Therapeutics that induce fiber type transitions may result in functional trade-offs (e.g., resistance to fatigue at the expense of maximal force production, or vice versa). This has been demonstrated in myostatin KO mice where selective increase in type II fibers at the expense of type I is associated with reduced endurance capacity

[74] and accelerated fatigue in isolated muscles [75]. Similarly, post-developmental KO of myostatin was associated with a significant reduction in voluntary wheel running and wild-type adult mice administered an AAV expressing the myostatin propeptide to antagonize myostatin activity presented with reduced forced distance running and greater fatigability in isolated muscles [76]. To date, a similar analysis in humans treated with muscle anabolics has not been completed but some insights may be garnered from examples of selective fiber type loss in the context of muscle atrophy. For example, elderly individuals with sarcopenia exhibit a selective atrophy of type II fibers [77–79], which has been shown to be associated with fracture related falls that may be attributed to inability to generate sufficient force needed to overcome loss of balance [77,79]. In contrast, heart failure patients present with a slow to fast fiber type transition that is associated with severe intolerance to physical exercise and accelerated fatigue during normal activities of daily living [80,81]. Thus, a muscle anabolic that preserves the pre-treatment fiber type profile is likely to benefit the patient by maintaining energy to perform normal activities of daily living while retaining the force generating capacity to prevent falls and injuries.

In the current study, ACE-031 administration was associated with a significant increase in both absolute and specific force of marmoset EDL muscles. In myostatin deficient mice, reports have shown moderate or no improvement in absolute force and a significant reduction in specific force [75,82]; an indication that improvements in skeletal muscle contractility do not occur proportionately to increases in skeletal muscle mass in myostatin-deficient mice. On the other hand, pharmacological inhibition of myostatin has been shown to increase absolute force of the EDL [18,23,24,57] and soleus [83] and Bogdanovich et al. reported a significant increase in specific force [24] in myostatin propeptide-mediated blockade. Consistent with ACE-031 administration, two groups, Minetti et al. [84] and Loro et al. [85] independently observed an increase in both absolute and specific force in dystrophic mice treated with trichostatin, a deacetylase inhibitor. The principal effector of deacetylase inhibitors in skeletal muscle is follistatin [86], which has been shown to inhibit myostatin as well as other negative regulators of skeletal muscle [12,87,88]. These observations further support the concept that a more broad-based therapeutic strategy, targeting multiple ActRIIB ligands, may be particularly beneficial in concurrently augmenting muscle mass and force production. Our findings indicate that ACE-031 is associated with greater lean body mass and muscle fiber cross-sectional area in a non-fiber-type specific manner. These skeletal muscle morphological effects translate to greater strength and functional capacity, manifested as an increase in both absolute and specific force. Consequently, this study confirms that concomitantly addressing multiple inhibitors of muscle growth, e.g., myostatin in combination with activins, through distinct approaches (antibody vs. ligand trap), might be more advantageous than targeting them independently. Assessing the impact of the dual-targeted pathways (myostatin and activins) in clinical trials resulted in much better changes in the body composition (increase lean body mass and reduction of fat mass), and in bone health [40,89] versus targeting myostatin alone [90]. Therefore, given the progressive muscle wasting nature of various human dystrophies and myopathies, a therapeutic strategy targeting multiple regulatory pathways of skeletal muscle could be more beneficial.

A limitation of this study is that the relatively small sample size, particularly in the control group (n = 4; 2 male, 2 female) compared with the treated group (n = 8; 4 male, 4 female), may have resulted in inadequate statistical power to detect subtle but potentially meaningful treatment effects. This limitation was underscored by a vehicle treated animal that showed an unexpected increase in body weight and muscle mass over the course of the study—an outcome anticipated in the ACE-031 treated group—which likely contributed to variability, limiting the ability to detect group differences. The authors concede that this limitation should be taken into account when interpreting the findings and their generalizability. An additional limitation was the use of the biceps brachii muscle for fiber typing and fiber CSA analysis and the use of the EDL muscle for functional analysis, which does not allow us to directly conclude on structure and function in the same muscle. The use of the EDL for functional measures was based on anatomical and methodological issues. The EDL muscle is a fusiform muscle, in which the muscle fibers run parallel to each other, with no pennation angle, and tapers at proximal and distal myotendinous junctions. Surgically, the EDL can be dissected with the proximal and distal tendons intact, leaving the muscle undamaged for subsequent functional assessment. This makes the EDL muscle ideal for functional studies as isometric force is directed in one direction between the connections to the force transducer and muscle stimulation

chamber. Although the histological data in the biceps brachii and the functional data in the EDL are supportive in this study, systemic therapies in general may not produce similar effects in every muscle.

Despite the potential of ActRIIB ligand traps to ameliorate muscle wasting, the experience from the ACE-031 clinical program underscores important safety considerations for this drug class. ACE-031 was discontinued in a Phase II trial in Duchenne muscular dystrophy due to vascular-related adverse events, including epistaxis, telangiectasia, and gum bleeding, which were later attributed to inhibition of ligands beyond myostatin and activin A, particularly BMP9 [91], which plays a critical role in vascular homeostasis. This discovery highlights that broad inhibition of TGF-β family ligands can produce effects that extend beyond muscle. Thus, while ActRIIB-targeted therapies remain a promising approach to increasing muscle mass and function, the ACE-031 findings emphasize the need for more selective ligand targeting and the inclusion of monitoring of vascular endpoints in future clinical trials. Indeed, KER-065, an ActRIIA and ActRIIB chimera ligand trap targeting myostatin and activins recently completed phase I clinical trials, and has already demonstrated an ability to stimulate significant improvements in muscle mass and strength, and bone health in animal models [92–94]. Moreover, KER-065 was shown to have a 400-fold lower inhibitory activity against BMP9 [93]. As part of the phase I trial monitoring for epistaxis and telangiectasis was included and none reported in healthy volunteers [94]. Overall, the potential of dual inhibition of myostatin and activin pathways presents a compelling direction for future research and clinical applications in addressing muscle-wasting conditions. Further human studies are warranted to explore this therapeutic strategy and its implications for safely improving patient outcomes.

## Supporting information

**S1 Table. Serum chemistry panels.**
(XLSX)

**S2 Table. ACE-031 pharmacokinetic analysis.**
(XLSX)

**S1 File. Data used for statistical analyses.**
(XLSX)

## Acknowledgments

We would like to thank Tim Ahern for his help in editing this manuscript.

## Author contributions

**Conceptualization:** Jasbir Seehra.

**Data curation:** Samuel M. Cadena, Jennifer L. Lachey.

**Formal analysis:** Samuel M. Cadena, Sasha Bogdanovich, R. Scott Pearsall, Elizabeth Curran, Ryan Faucette, Joan Lane, Jennifer L. Lachey, Emidio E. Pistilli.

**Investigation:** Jasbir Seehra, Emidio E. Pistilli.

**Methodology:** Samuel M. Cadena, Sasha Bogdanovich, Abigail Pullen, R. Scott Pearsall, Elizabeth Curran, Ryan Faucette, Joan Lane, Jasbir Seehra, Jennifer L. Lachey, Emidio E. Pistilli.

**Supervision:** Tejvir S. Khurana, Jasbir Seehra.

**Visualization:** Samuel M. Cadena, Sasha Bogdanovich, Tejvir S. Khurana, Alan D. Mizener, Emidio E. Pistilli.

**Writing – original draft:** Samuel M. Cadena, Sasha Bogdanovich, Tejvir S. Khurana, Abigail Pullen, R. Scott Pearsall, Elizabeth Curran, Ryan Faucette, Joan Lane, Jasbir Seehra, Jennifer L. Lachey, Emidio E. Pistilli.

**Writing – review & editing:** Samuel M. Cadena, Sasha Bogdanovich, Tejvir S. Khurana, Abigail Pullen, R. Scott Pearsall, Elizabeth Curran, Ryan Faucette, Joan Lane, Jasbir Seehra, Jennifer L. Lachey, Alan D. Mizener, Emidio E. Pistilli.

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
