## [Decision Letter · Decision Letter 0]

30 Oct 2025

Dear Dr. Pistilli,

Thank you for submitting your manuscript to PLOS ONE. After careful consideration, we feel that it has merit but does not fully meet PLOS ONE’s publication criteria as it currently stands. Therefore, we invite you to submit a revised version of the manuscript that addresses the points raised during the review process.

We look forward to receiving your revised manuscript.

Kind regards,

Keisuke Hitachi

Academic Editor

PLOS ONE

Journal Requirements:

We would like to thank Tim Ahern for his help in editing this manuscript. E. Pistilli acknowledges funding through the National Institute of Arthritis Musculoskeletal and Skin diseases (NIAMS; AR079445).

I have read the journal's policy and the authors have declared no competing interests. Co-authors were former employees of a company Acceleron that holds the patent to the drug. However, the company is no longer active.

6. We note that your Data Availability Statement is currently as follows: All relevant data are within the manuscript and its Supporting Information files.

7. Please amend the manuscript submission data (via Edit Submission) to include author Tejvir S. Khurana

8. Please amend your authorship list in your manuscript file to include author Tejvhir Khurana

9. We note that you have included the phrase “data not shown” in your manuscript. Unfortunately, this does not meet our data sharing requirements. PLOS does not permit references to inaccessible data. We require that authors provide all relevant data within the paper, Supporting Information files, or in an acceptable, public repository. Please add a citation to support this phrase or upload the data that corresponds with these findings to a stable repository (such as Figshare or Dryad) and provide and URLs, DOIs, or accession numbers that may be used to access these data. Or, if the data are not a core part of the research being presented in your study, we ask that you remove the phrase that refers to these data.

10. Please include captions for your Supporting Information files at the end of your manuscript, and update any in-text citations to match accordingly. Please see our Supporting Information guidelines for more information: http://journals.plos.org/plosone/s/supporting-information.

Reviewers' comments:

Reviewer's Responses to Questions

**Comments to the Author**

1. Is the manuscript technically sound, and do the data support the conclusions?

Reviewer #1: Partly

Reviewer #2: Partly

2. Has the statistical analysis been performed appropriately and rigorously?

Reviewer #1: N/A

Reviewer #2: No

3. Have the authors made all data underlying the findings in their manuscript fully available?

Reviewer #1: Yes

Reviewer #2: Yes

4. Is the manuscript presented in an intelligible fashion and written in standard English?

Reviewer #1: Yes

Reviewer #2: Yes

Reviewer #1: This manuscript presents a well-conducted preclinical study investigating the effects of ACE-031, a soluble ActRIIB receptor, on muscle mass and function in the common marmoset, a non-human primate (NHP) model. The study is timely and addresses a critical gap in the field of myostatin/activin pathway inhibition, namely, the translation of promising murine data to a more clinically relevant primate model. The experimental design is robust, incorporating longitudinal body composition analysis, terminal morphometric assessment of muscle fibers, and functional ex vivo contractility measurements. The results clearly demonstrate that ACE-031 treatment increases lean mass, muscle fiber cross-sectional area, and muscle strength in marmosets.

While the findings are positive and support the therapeutic potential of broad ActRIIB ligand inhibition, the manuscript has significant weaknesses in its statistical reporting, data interpretation, and discussion that must be addressed prior to publication.

The study uses a control group of n=4 and a treatment group of n=8. There is no justification provided for this sample size, and no power calculation is reported. With such a small control group (n=4), the study is severely underpowered. This increases the risk of both Type I and Type II errors. The authors themselves note that one outlier in the control group (Marmoset "A") may have affected the group means and statistical outcomes. A small n amplifies the impact of such outliers.

The abstract states, "Marmosets administered ACE-031 showed a significant increase in body weight and lean body mass from baseline, while no change was seen in the vehicle-treated controls." However, the statistical analysis in the results (Fig 1A, B) reports a main effect of time, but no main effect of treatment and no significant interaction for body weight. For lean mass, there is a significant interaction but no main effect of treatment. This indicates that while the ACE-031 group increased over its own baseline, the statistical comparison between the treatment and control groups over time is not robust. The phrasing in the abstract is therefore misleading and must be corrected to accurately reflect the statistical findings.

The use of one-tailed t-tests is questionable and requires strong justification. One-tailed tests should only be used when there is a priori, directional hypothesis with no possibility of an effect in the opposite direction. While the hypothesis was for an increase, the use of one-tailed tests doubles the risk of a false positive. The journal's policy on this should be checked, and it is standard practice to use two-tailed tests unless there is an overwhelming justification otherwise. All analyses using one-tailed tests should be re-run with two-tailed tests, or a compelling justification must be provided.

The discussion reads like a positive interpretation of the data without a balanced critical perspective. It fails to adequately address the central paradox of the ACE-031 clinical program: why did a drug that shows efficacy in NHPs (this study) and humans (increased lean mass) fail in Phase II trials for Duchenne Muscular Dystrophy (DMD) due to safety concerns (epistaxis, telangiectasias)?

The discussion on fiber types is informative but could be more incisive. The authors conclude that the inherent metabolic profile is unaltered, which is a strength. However, they could more directly discuss the potential clinical implications of hypertrophy in both fiber types versus a selective effect.

Reviewer #2: This paper studies the effects of ACE-031, a protein that blocks activin receptor type IIB ligands, on muscle growth and strength in common marmosets. The topic is important and relevant, especially for developing new treatments for muscle-wasting diseases. The study is well organized and the writing is clear. Using a non-human primate model makes the results more meaningful for human application.

However, there are some problems with the experimental design, data analysis, and interpretation that should be fixed before the paper can be published.

1. The goal of testing ACE-031 in primates is reasonable, since mouse models don’t always match human biology. Still, the novelty is limited because ACE-031 and other ActRIIB blockers have been studied before. The authors should explain what new information this study adds. For example, what unique features of the marmoset model make these results important?

2. Only 12 animals were used (8 treated, 4 controls). This small sample size reduces the strength of the findings. Also, using one-tailed t-tests is not appropriate here, since the direction of effect was not fully predictable. Use two-tailed tests, include a power calculation, and explain how the small sample size might affect the results.

3. The results show increases in lean mass and body weight, but some effects were not significant when using two-way ANOVA. One animal in the control group gained a lot of weight, which likely affected group averages. Re-analyze the data with and without that animal and discuss how it changes the outcome. Show individual data points in the figures to make variability clear.

4. The study measured strength in the EDL muscle, but muscle growth was mainly shown in the biceps. This makes it hard to connect the structural and functional results. Explain why EDL was used instead of biceps, or include strength data from a muscle that also showed hypertrophy.

5. The study does not report much about safety. Since ACE-031 has been linked to bleeding problems in past human studies, this part is important. Include data or at least a summary on blood tests, organ changes, or any visible side effects.

6. The Discussion repeats a lot from previous studies and doesn’t say much about what these specific results mean. It would help to explain the possible biological reasons for the observed effects, such as which muscle pathways might be involved. Add more explanation about how ACE-031 might cause muscle growth and strength increases, and compare how this may differ from mice or humans.

7. Use consistent terms for “lean mass,” “muscle mass,” and “lean body mass.”

8. State if the data were tested for normality before using parametric tests.

After these revisions, the study could make a good contribution to the field.

**Do you want your identity to be public for this peer review?** For information about this choice, including consent withdrawal, please see our Privacy Policy

Reviewer #1: No

Reviewer #2: No

---

## [Author Response · Author response to Decision Letter 1]

13 Dec 2025

DATE: December 14, 2025

RE: Revision of Manuscript #PONE-D-25-54403

Response to Reviewer Comments (PONE-D-25-54403)

The Authors want to thank the Editor and Expert Reviewers for the thoughtful and thorough review of our submitted manuscript “ACE-031, a Soluble Activin Type IIB Receptor, Increases Muscle Mass and Strength in the Common Marmoset (Callithrix jacchus).” We have carefully considered all the suggestions and revised the manuscript accordingly. We have also addressed the additional requirements required for publication in PLOS ONE, which has strengthened our manuscript considerably. A point-by-point response to each Reviewer suggestion is presented below, which we hope sufficiently addressed all Reviewer concerns.

Reviewer #1:

1) This manuscript presents a well-conducted preclinical study investigating the effects of ACE-031, a soluble ActRIIB receptor, on muscle mass and function in the common marmoset, a non-human primate (NHP) model. The study is timely and addresses a critical gap in the field of myostatin/activin pathway inhibition, namely, the translation of promising murine data to a more clinically relevant primate model. The experimental design is robust, incorporating longitudinal body composition analysis, terminal morphometric assessment of muscle fibers, and functional ex vivo contractility measurements. The results clearly demonstrate that ACE-031 treatment increases lean mass, muscle fiber cross-sectional area, and muscle strength in marmosets.

Response: Thank you.

2) While the findings are positive and support the therapeutic potential of broad ActRIIB ligand inhibition, the manuscript has significant weaknesses in its statistical reporting, data interpretation, and discussion that must be addressed prior to publication.

Response: We have revised the manuscript to address the identified weaknesses in statistical reporting, data interpretation and discussion.

3) The study uses a control group of n=4 and a treatment group of n=8. There is no justification provided for this sample size, and no power calculation is reported. With such a small control group (n=4), the study is severely underpowered. This increases the risk of both Type I and Type II errors. The authors themselves note that one outlier in the control group (Marmoset "A") may have affected the group means and statistical outcomes. A small n amplifies the impact of such outliers.

Response: We agree with the Reviewer and concede that this was indeed a limitation of the study. We have included the following paragraph in the Discussion to identify this issue:

“A limitation of this study is that the relatively small sample size, particularly in the control group (n = 4; 2 male, 2 female) compared with the treated group (n = 8; 4 male, 4 female), may have resulted in inadequate statistical power to detect subtle but potentially meaningful treatment effects. This limitation was underscored by a vehicle treated animal that showed an unexpected increase in body weight and muscle mass over the course of the study—an outcome anticipated in the ACE-031 treated group—which likely contributed to variability, limiting the ability to detect group differences. The authors concede that this limitation should be taken into account when interpreting the findings and their generalizability.”

We hope this addition to the Discussion addresses the Reviewer’s concern.

4) The abstract states, "Marmosets administered ACE-031 showed a significant increase in body weight and lean body mass from baseline, while no change was seen in the vehicle-treated controls." However, the statistical analysis in the results (Fig 1A, B) reports a main effect of time, but no main effect of treatment and no significant interaction for body weight. For lean mass, there is a significant interaction but no main effect of treatment. This indicates that while the ACE-031 group increased over its own baseline, the statistical comparison between the treatment and control groups over time is not robust. The phrasing in the abstract is therefore misleading and must be corrected to accurately reflect the statistical findings.

Response: We have revised the wording in the Abstract to more accurately reflect the statistical findings.

5) The use of one-tailed t-tests is questionable and requires strong justification. One-tailed tests should only be used when there is a priori, directional hypothesis with no possibility of an effect in the opposite direction. While the hypothesis was for an increase, the use of one-tailed tests doubles the risk of a false positive. The journal's policy on this should be checked, and it is standard practice to use two-tailed tests unless there is an overwhelming justification otherwise. All analyses using one-tailed tests should be re-run with two-tailed tests, or a compelling justification must be provided.

Response: The use of the one-tailed t-test was in response to the large body of literature indicating that myostatin and ActRIIB inhibition, either through genetic or pharmacologic strategies, was associated with greater skeletal muscle mass and strength [1-14]. We felt that the one-tailed t-test was justified, as we expected to see a similar response in the NHP marmoset model. However, given the uncertainty of moving from murine models to NHP models, the Reviewers concern, and PLOS ONE’s policy, we re-analyzed the data using a two-tailed t-test. The majority of the parameters analyzed by t-test remained unchanged with regards to statistical significance. The one parameter that was changed from significant to not significant was the EDL absolute twitch force: one-tailed t-test p=0.0266; two-tailed t-test p=0.0533. We have revised the manuscript to reflect this change in statistical analysis, including revised text in the Materials and methods and Results sections and revised Figure 3. We thank the Reviewer for this suggestion and hope that the revised statistical analysis strengthens the conclusions of the data.

6) The discussion reads like a positive interpretation of the data without a balanced critical perspective. It fails to adequately address the central paradox of the ACE-031 clinical program: why did a drug that shows efficacy in NHPs (this study) and humans (increased lean mass) fail in Phase II trials for Duchenne Muscular Dystrophy (DMD) due to safety concerns (epistaxis, telangiectasias)?

Response: We acknowledge that this is indeed a potential safety concern with this class of drug. We’ve added more to this point in the Discussion:

“Despite the potential of ActRIIB ligand traps to ameliorate muscle wasting, the experience from the ACE-031 clinical program underscores important safety considerations for this drug class. ACE-031 was discontinued in a Phase II trial in Duchenne muscular dystrophy due to vascular-related adverse events, including epistaxis, telangiectasia, and gum bleeding, which were later attributed to inhibition of ligands beyond myostatin and activin A, particularly BMP9[15], which plays a critical role in vascular homeostasis. This discovery highlights that broad inhibition of TGF-β family ligands can produce effects that extend beyond muscle. Thus, while ActRIIB-targeted therapies remain a promising approach to increasing muscle mass and function, the ACE-031 findings emphasize the need for more selective ligand targeting and the inclusion of monitoring of vascular endpoints in future clinical trials. Indeed, RKER-06...

7) The discussion on fiber types is informative but could be more incisive. The authors conclude that the inherent metabolic profile is unaltered, which is a strength. However, they could more directly discuss the potential clinical implications of hypertrophy in both fiber types versus a selective effect.

Response: Thank you for the suggestion. We agree that this is an important aspect of our treatment paradigm the deserves a more detailed discussion. The Discussion has been edited to include the following:

“Slow‐twitch fibers maintain oxidative capacity and fatigue resistance, supporting posture and endurance-based tasks, whereas fast‐twitch fibers provide high power output and rapid contractile kinetics needed for short bursts of energy. Therapeutics that induce fiber type transitions may result in functional trade-offs (e.g., resistance to fatigue at the expense of maximal force production, or vice versa). This has been demonstrated in myostatin KO mice where selective increase in type II fibers at the expense of type I is associated with reduced endurance capacity [16] and accelerated fatigue in isolated muscles[1]. Similarly, post-developmental KO of myostatin was associated with a significant reduction in voluntary wheel running and wild-type adult mice administered an AAV expressing the myostatin propeptide to antagonize myostatin activity presented with reduced forced distance running and greater fatigability in isolated muscles [17]. To date, a similar analysis in humans treated with muscle anabolics has not been completed but some insights may be garnered from examples of selective fiber type loss in the context of muscle atrophy. For example, elderly individuals with sarcopenia exhibit a selective atrophy of type II fibers [18-20], which has been shown to be associated with fracture related falls that may be attributed to inability to generate sufficient force needed to overcome loss of balance[18, 20]. In contrast, heart failure patients present with a slow to fast fiber type transition that is associated with severe intolerance to physical exercise and accelerated fatigue during normal activities of daily living [21, 22]. Thus, a muscle anabolic that preserves the pre-treatment fiber type profile is likely to benefit the patient by maintaining energy to perform normal activities of daily living while retaining the force generating capacity to prevent falls and injuries.”

Reviewer #2:

1) This paper studies the effects of ACE-031, a protein that blocks activin receptor type IIB ligands, on muscle growth and strength in common marmosets. The topic is important and relevant, especially for developing new treatments for muscle-wasting diseases. The study is well organized, and the writing is clear. Using a non-human primate model makes the results more meaningful for human application. However, there are some problems with the experimental design, data analysis, and interpretation that should be fixed before the paper can be published.

Response: Thank you for the positive words on our manuscript. We have revised the manuscript to address your concerns on the experimental design, data analysis and interpretation.

2) The goal of testing ACE-031 in primates is reasonable, since mouse models don’t always match human biology. Still, the novelty is limited because ACE-031 and other ActRIIB blockers have been studied before. The authors should explain what new information this study adds. For example, what unique features of the marmoset model make these results important? Emphasize publications that show NHP to be superior to rodents when evaluating ACTRIIB ligand traps. E.g., different levels of circulating ligands in rodents vs NHP/humans.

Response: Thank you for the suggestion. This was briefly mentioned in the Introduction but is worthwhile to revisit and further expand upon in the Discussion. We have now included the following in the Discussion:

“Activin A levels in monkeys and humans are 3-4 fold higher than mice and rats [23] and myostatin levels in mice are more than 10-fold higher than humans[24] suggesting that myostatin specific treatments that appear promising in mice are not as likely to produce clinically meaningful outcomes in humans. Indeed, although myostatin specific treatments have been shown to be efficacious in a mouse model of muscular dystrophy [2, 3, 25], selective antagonism of myostatin has failed to show meaningful clinical benefit in patients with muscular dystrophies [26-28]. Further, a systematic analysis of the pharmacokinetic and pharmacodynamic profile of a myostatin neutralizing antibody in mice, rats, monkeys and humans suggest that the amount of drug needed to induce comparable increases in muscle mass in monkeys was 20-fold higher than what is needed in mice; indicating substantial species differences in sensitivity to myostatin specific treatments[29]. Thus, the use of NHPs is likely to be a superior predictor of efficacy for treatments targeting myostatin and activin A in humans.”

We hope this added text sufficiently addresses the Reviewer’s concerns.

3) Only 12 animals were used (8 treated, 4 controls). This small sample size reduces the strength of the findings. Also, using one-tailed t-tests is not appropriate here, since the direction of effect was not fully predictable. Use two-tailed tests, include a power calculation, and explain how the small sample size might affect the results.

Response: Your concerns regarding the sample size and the use of a one-tailed t-test were shared with Reviewer 1. We have added the following text to the Discussion regarding the same sizes:

“A limitation of this study is that the relatively small sample size, particularly in the control group (n = 4; 2 male, 2 female) compared with the treated group (n = 8; 4 male, 4 female), may have resulted in inadequate statistical power to detect subtle but potentially meaningful treatment effects. This limitation was underscored by a vehicle treated animal that showed an unexpected increase in body weight and muscle mass over the course of the study—an outcome anticipated in the ACE-031 treated group—which likely contributed to variability, limiting the ability to detect group differences. The authors concede that this limitation should be taken into account when interpreting the findings and their generalizability.”

Regarding the use of a one-tailed t-test, this was in response to the large body of literature indicating that myostatin and ActRIIB inhibition, either through genetic or pharmacologic strategies, was associated with greater skeletal muscle mass and strength [1-14]. We felt that the one-tailed t-test was justified, as we expected to see a similar response in the NHP marmoset model. However, given the uncertainty of moving from murine models to NHP models, the Reviewers concern, and PLOS ONE’s policy, we re-analyzed the data using a two-tailed t-test. The majority of the parameters analyzed by t-test remained unchanged with regards to statistical significance. The one parameter that was changed from significant to not significant was the EDL absolute twitch force: one-tailed t-test p=0.0266; two-tailed t-test p=0.0533. We have revised the manuscript to reflect this change in statistical analysis, including revised text in the Materials and methods and Results sections and revised Figure 3. We thank the Reviewer for these suggestions and hope that the revisions strengthen the conclusions of the data.

4) The results show increases in lean mass and body weight, but some effects were not significant when using two-way ANOVA. One animal in the control group gained a lot of weight, which likely affected group averages. Re-analyze the data with and without that animal and discuss how it changes the outcome. Show individual data points in the figures to make variability clear.

Response: We agree with the Reviewer that this one control marmoset contributes to variability in our data. We have performed the statistical analysis as suggested by the Reviewer. Here we provide a comparison of the statistical analysis for body weight and lean body mass with and without that one specific marmoset. As seen in the analysis, the statistical significance does not change when removing the data points associated with this marmoset. For this reason, we feel it is better to maintain the original analysis that includes this marmoset.

We have also provided here new graphs for the body weight, lean body mass and fat mass that show data points for each individual marmoset.

These graphs become quite busy in this format. It is our preference to utilize the figures contained within our original submission. However, if the Reviewer feels these new graphs better present the variability in the data an

---

## [Decision Letter · Decision Letter 1]

26 Dec 2025

Dear Dr. Pistilli,

Thank you for submitting your manuscript to PLOS ONE. After careful consideration, we feel that it has merit but does not fully meet PLOS ONE’s publication criteria as it currently stands. Therefore, we invite you to submit a revised version of the manuscript that addresses the points raised during the review process.

We look forward to receiving your revised manuscript.

Kind regards,

Keisuke Hitachi

Academic Editor

PLOS One

Journal Requirements:

Additional Editor Comments:

Thank you for addressing the comments. However, the reviewers have pointed out that several concerns remain. I also feel that the reviewers' comments are well-founded, so please address them appropriately. Please pay particular attention to the statistics and the differences in the analyzed muscles.

Reviewer's Responses to Questions

**Comments to the Author**

Reviewer #2: (No Response)

2. Is the manuscript technically sound, and do the data support the conclusions?

Reviewer #2: Yes

3. Has the statistical analysis been performed appropriately and rigorously?

Reviewer #2: Yes

4. Have the authors made all data underlying the findings in their manuscript fully available?

Reviewer #2: Yes

5. Is the manuscript presented in an intelligible fashion and written in standard English?

Reviewer #2: Yes

Reviewer #2: Thank you for the revised manuscript and the detailed responses. The authors have made clear improvements, and many of the earlier concerns are now addressed. The paper reads better, the analyses are more transparent, and the Discussion has more depth. I appreciate the effort put into the revisions.

Below are my updated comments.

1. Statistical clarity

The authors corrected the statistical approach and re-ran the tests as two-tailed, which is appropriate. This fixes one of the main issues from the previous review. However, the interpretation of the mixed-model results is still a bit difficult for readers who are not familiar with these analyses. In the current form, the Results section may still leave some confusion.

Add 1–2 short sentences explaining, in simple terms, what the significant Time × Treatment interaction means in practice (e.g., lean mass increased over time in the treated group but stayed stable in controls). This will make the results easier to understand.

2. Muscle function interpretation

The additional explanation on force measurements is helpful. Still, there is a structural–functional gap:

• hypertrophy was measured mainly in the biceps

• force was measured in the EDL

The authors justify this choice scientifically, but a brief reminder in the Discussion that systemic therapies may not produce identical effects in every muscle would help readers understand this limitation.

3. Some figure legends still have small grammatical issues and could be proofread once more.

4. Use consistent terminology for lean mass throughout the paper.

5. Ensure that all figure labels match the panels exactly.

The manuscript is clearly improved. The authors addressed most of the earlier concerns in a meaningful way, and the scientific content is now stronger and more transparent. A few remaining issues relate to clarity and presentation rather than major flaws.

**Do you want your identity to be public for this peer review?** For information about this choice, including consent withdrawal, please see our Privacy Policy

Reviewer #2: No

---

## [Author Response · Author response to Decision Letter 2]

7 Jan 2026

DATE: January 5, 2025

RE: Revision of Manuscript #PONE-D-25-54403R1

Response to Reviewer Comments (PONE-D-25-54403R1)

The Authors want to thank the Editor and Expert Reviewers for the thoughtful and thorough review of our revised submitted manuscript “ACE-031, a Soluble Activin Type IIB Receptor, Increases Muscle Mass and Strength in the Common Marmoset (Callithrix jacchus).” A point-by-point response to the Reviewer suggestions is presented below.

Reviewer #2: Thank you for the revised manuscript and the detailed responses. The authors have made clear improvements, and many of the earlier concerns are now addressed. The paper reads better, the analyses are more transparent, and the Discussion has more depth. I appreciate the effort put into the revisions. Below are my updated comments.

Response: Thank you.

1. Statistical clarity

The authors corrected the statistical approach and re-ran the tests as two-tailed, which is appropriate. This fixes one of the main issues from the previous review. However, the interpretation of the mixed-model results is still a bit difficult for readers who are not familiar with these analyses. In the current form, the Results section may still leave some confusion.

Add 1–2 short sentences explaining, in simple terms, what the significant Time × Treatment interaction means in practice (e.g., lean mass increased over time in the treated group but stayed stable in controls). This will make the results easier to understand.

Response: Thank you for this suggestion. We have added a sentence to the Results section to clarify the mixed models results.

2. Muscle function interpretation

The additional explanation on force measurements is helpful. Still, there is a structural–functional gap:

• hypertrophy was measured mainly in the biceps

• force was measured in the EDL

The authors justify this choice scientifically, but a brief reminder in the Discussion that systemic therapies may not produce identical effects in every muscle would help readers understand this limitation.

Response: We have added text to the end of the Discussion to address this limitation. Specifically, we added in the rationale for the use of the EDL for functional assessment, and then added the text suggested by the Reviewer regarding the potential that systemic therapies may not similarly affect all muscles.

3. Some figure legends still have small grammatical issues and could be proofread once more.

Response: Thank you for this and we apologize for that overcite. The figure legends have been corrected and should not contain any additional grammatical errors.

4. Use consistent terminology for lean mass throughout the paper.

Response: Thank you. We have used “lean body mass” throughout the manuscript.

5. Ensure that all figure labels match the panels exactly.

Response: We have checked figure labels and the figures to ensure they match.

The manuscript is clearly improved. The authors addressed most of the earlier concerns in a meaningful way, and the scientific content is now stronger and more transparent. A few remaining issues relate to clarity and presentation rather than major flaws.

Response: Thank you once again for the additional suggestions and corrections to our manuscript.

---

## [Decision Letter · Decision Letter 2]

27 Jan 2026

ACE-031, a Soluble Activin Type IIB Receptor, Increases Muscle Mass and Strength in the Common Marmoset (Callithrix jacchus)

PONE-D-25-54403R2

Dear Dr. Pistilli,

We’re pleased to inform you that your manuscript has been judged scientifically suitable for publication and will be formally accepted for publication once it meets all outstanding technical requirements.

Kind regards,

Keisuke Hitachi

Academic Editor

PLOS One

Additional Editor Comments (optional):

Reviewers' comments:

Reviewer's Responses to Questions

**Comments to the Author**

Reviewer #2: All comments have been addressed

2. Is the manuscript technically sound, and do the data support the conclusions?

Reviewer #2: Yes

3. Has the statistical analysis been performed appropriately and rigorously?

Reviewer #2: Yes

4. Have the authors made all data underlying the findings in their manuscript fully available?

Reviewer #2: Yes

5. Is the manuscript presented in an intelligible fashion and written in standard English?

Reviewer #2: Yes

Reviewer #2: I have reviewed the latest revised version of the manuscript as well as the authors’ detailed responses. The authors have addressed all previous comments thoroughly and thoughtfully. The manuscript is now clear, well organized, and scientifically sound.

The statistical explanations have been clarified, the Discussion flows more naturally, terminology is consistent, and the figures and legends have been corrected. The additional notes regarding muscle selection for functional testing and the potential variability between muscles in systemic treatments are helpful and complete the interpretation.

I do not see any remaining issues that require further revision. The study is well designed, the data are presented transparently, and the conclusions are supported by the results. The manuscript is suitable for publication in its current form.

**Do you want your identity to be public for this peer review?** For information about this choice, including consent withdrawal, please see our Privacy Policy

Reviewer #2: No

---

## [Editor Report · Acceptance letter]

PONE-D-25-54403R2

PLOS One

Dear Dr. Pistilli,

I'm pleased to inform you that your manuscript has been deemed suitable for publication in PLOS One. Congratulations! Your manuscript is now being handed over to our production team.

Kind regards,

on behalf of

Dr. Keisuke Hitachi

Academic Editor

PLOS One